# Instrumentation for Verification of Shunt Active Power Filter Algorithms

**DOI:** 10.3390/s23208494

**Published:** 2023-10-16

**Authors:** Jan Baros, Petr Bilik, Rene Jaros, Lukas Danys, Jan Strossa, Lukas Hlavaty, Radek Martinek

**Affiliations:** 1Department of Cybernetics and Biomedical Engineering, VSB–Technical University of Ostrava, 17. Listopadu 15, 708 33 Ostrava, Czech Republic; 2Department of Electronics, VSB–Technical University of Ostrava, 17. Listopadu 15, 708 33 Ostrava, Czech Republic

**Keywords:** SAPF, LabVIEW, Notch–LMS, Notch–RLS, virtual instrumentation

## Abstract

This article presents a comprehensive system for testing and verifying shunt active power filter control methods. The aim of this experimental platform is to provide tools to a user to objectively compare the individual control methods. The functionality of the system was verified on a hardware platform using least mean squares and recursive least squares algorithms. In the experiments, an average relative suppression of the total harmonic distortion of 22% was achieved. This article describes the principle of the shunt active power filter, the used experimental platform of the controlled current injection source, its control system based on virtual instrumentation and control software and ends with experimental verification. The discussion of the paper outlines the extension of the experimental platform with the cRIO RTOS control system to reduce the latency of reference current generation and further planned research including motivation.

## 1. Introduction

As a result of massive modernization and constant human expansion, there is also a massive expansion of consumer electronics and electrical appliances with power electronic circuits [1]. These appliances typically improve the quality of life and contribute to overall well-being of humankind. However, these benefits are also associated with some unpleasant negatives—the massive use of power electronics contributes to a deterioration of the power quality (PQ) in the distribution system and thus to the quality of the electricity that is distributed and sold to customers. The most common type of load that degrades PQ is non-linear load, which draws a current from the grid that contains a large number of unwanted higher harmonic components.

These harmonic components cause disturbances in the distribution network (outages, undervoltage, overvoltage) and affect the magnitude of the harmonic distortion of the voltage. Since appliances are sensitive to changes in the magnitude and shape of voltage, their lifetime is reduced and service and maintenance costs increase drastically [2,3]. Other problems caused by harmonic components of the current drawn by appliances are low power factor in the distribution network, unwanted acoustic sounds, current spikes or overheating and overloading of conductors in the power network [4,5]. In order to avoid these problems and the damage associated with the considerable financial costs, companies are investing considerable resources in researching ways to reduce and prevent harmonic current components. Harmonically distorted currents can be eliminated by using power filters, which may have passive [6,7], active [8] or hybrid [9,10,11] characteristics.

Active power filters are not as widely used as passive filters due to their higher cost. Therefore, passive filters are deployed for the time being—they are made for a specific case and are deployed on a given part of the distribution network. The main advantages of passive filters are simplicity of design and low deployment costs. The disadvantages are the large size, the problematic debugging resulting in inefficient filtration due to degradation of components and, most importantly, the inability to actively adapt to new states of the distribution network [6]. However, this is an insufficient solution in view of the massive expansion of power appliances in households and the expansion and automation of industrial production factories. To cover a wide range of disturbances, rapid filter adaptation and response to the current situation in the distribution network is required. These advantages are provided by the shunt active power filter (SAPF), which can be deployed where it is required the most [12].

A shunt active power filter (SAPF) is connected in parallel to the network and carries out the compensation by parallel injection or by draining harmonic current in/from the network. Non-linear load drains non-linear current $i_{L}\left(t\right)$ from the grid current i(t), which consists of fundamental harmonic ${\bar{i}}_{L}\left(t\right)$ and harmonic components ${\tilde{i}}_{L}\left(t\right)$ parts. The task of the SAPF is to precisely detect the higher harmonic part from the current signal and then inject it into power network with the opposite sign. As a result, the current drawn from the network has compensated harmonics and resembles an ideal sinusoidal waveform. Subsequently, only the fundamental harmonic power is drawn from the distribution network. The basic principle and block diagram of SAPF can be seen in Figure 1.

The control block of SAPF can be divided into four basic sections: harmonic current extraction (HCE), synchronization, DC voltage regulation and current generation. There are many different types of harmonic current extraction methods [14], grouped into several categories: time-domain-based [15,16,17,18,19,20,21,22,23], frequency-domain-based [14,24,25,26], adaptive algorithms [27,28,29,30], genetic-inspired algorithms and machine learning. The task of these methods is to perform mathematical separation of fundamental current and higher harmonic current. Harmonic current extracted by this separation is then used for active filtration. Main requirements for filtration methods are accuracy and rate of current generation. Further requirements for algorithms are the rate of convergence, stability and minimal mean squared error.

Multiple control methods of SAPF have been described in many publications over the years. Many of them were verified by real experiments on a real SAPF [21,27,31,32] or in a SW simulation environment [33,34,35,36,37]. Many of these methods were subjected to comparative experiments, but these experiments did not follow a uniform methodology. The conditions under which the methods were examined and evaluated differed from one publication to another. Therefore, it is not possible to objectively compare individual experiments of different authors [27]. However, the most frequently examined criterion when testing SAPF methods is the ability to suppress harmonic components, quantified in the form of total harmonic distortion of current (THD${}_{I}$). This parameter describes in percentage how much current waveform differs from ideal sinusoidal shape. Other often examined criteria are method convergence rate, implementation difficulty, ability to work under distorted voltage conditions, parameter configuration and the method’s suitability for single-phase or three-phase power networks.

Another prerequisite that needs to be examined is the objectivity towards the method itself in terms of the used SAPF HW. If an objective evaluation of the method is to be performed, it is necessary that the SAPF control unit does not exhibit a long latency, since it degrades the performance of the SAPF control method. Therefore, it is recommended to use very fast computational units, such as digital signal processors (DSPs) or a field programmable gate array (FPGA). Many authors used inappropriate HW in terms of determinism and latency. On the contrary, different authors used the FPGA [27,38,39], so again, direct comparison cannot be carried out.

For the reasons described above, our motivation was to develop an experimental platform which can evaluate the quality of harmonic extraction methods in different scenarios of power grid parameters using uniform criteria. The experimental platform was designed as an open architecture to access the SAPF problem and is easily modifiable and expandable. The control system is based on virtual instrumentation, which means that the software is responsible for the functionality of the whole platform. By using this approach, the changes in the explored area can be dynamically adapted, and the new SAPF control methods can be implemented and verified without HW adjustments. This allows older and newer methods to be implemented and compared based on uniform criteria.

This article presents a novel design and validation of a unique, custom-built experimental platform that allows evaluation of harmonic extraction methods in different power network parameter scenarios. Section 2 covers the HW & SW design of the presented experimental platform, describing its unique architecture. Section 3 describes the accuracy of the individual modules of the cDAQ control system, including the reason for performing the measurement chain adjustment and calibration. Section 4 discuss the methods used for instrumentation validation. Section 5 describes the early experiments used for system verification and discuss the results of these experiments. Section 6 discusses gaps in the design of the CICS platform and opportunities for improving the CICS system and outlines motivations and opportunities for further research. Lastly, Section 7 contains the evaluation of the achieved results.

## 2. Controllable Injection Current Source (CICS) Experimental Platform

In order to meet the requirements described in the introduction, it was necessary to design an experimental platform that allows an objective comparison of SAPF control methods [28]. A schematic of the proposed experimental platform can be seen in Figure 2.

The controllable injection current source (hereafter abbreviated as *CICS*) experimental platform consists of three injection current generators GEN 1–3 and three controllable load modules LOAD 1–3. Data acquisition and generation of the reference current (in the form of voltage shape used for CICS control analog input) is procured by the NI (Austin, TX, USA) *cDAQ-9189* system. Measured signals are signal conditioned before connection to A/D converters. A photo collage of the *CICS* experimental platform can be seen in Figure 3.

### 2.1. Three-Phase Compensation Current Generator

The *CICS* module itself is a one-phase-current controlled H-bridge converter that is connected to the grid via coupling inductance $L_{F}$. The *CICS* module is a single-phase programmable voltage source, which is phase-coupled to the grid voltage. The *CICS* module generates voltage shape $v_{G}\left(t\right)$ that is required to achieve the correct injection current $i_{G}\left(t\right)$ form. For simplicity, line inductance and resistance are not considered in following equation:(1)$$\begin{array}{c} v_{G}\left(t\right)=v_{S}\left(t\right)+L_{F}\frac{d\;i_{G}\left(t\right)}{d\;t}, \end{array}$$
where $v_{S}\left(t\right)$ is the distribution network voltage, $v_{G}\left(t\right)$ is the voltage of the generator, $L_{F}$ is the coupling inductance and the $i_{G}\left(t\right)$ is the injection current.

The maximum amplitude of current $i_{G}\left(t\right)$ that can be generated is limited to the difference between the distribution network voltage amplitude value $v_{S}\left(t\right)$ and the maximum value of generated voltage $v_{G}\left(t\right)$. The current $i_{G}\left(t\right)$ is also affected by the value of the coupling inductance $L_{F}$. Due to the inductance, the maximum amplitude of injection current $i_{G}\left(t\right)$ is frequency-dependent, and its value decreases with the increase in frequency. A simplified expression of the $I_{G}$ can be seen in Equations (Equation 2) and (Equation 3). Another task of inductance $L_{F}$ is to filter the current ripple caused by the high-frequency switching of the IGBT transistors inside the current source.

Due to the nature of the task that this CICS module performs, it is not possible for generator to compensate the fundamental $f_{1}$ harmonic of the current signal, but this is not the goal.
(2)$$\begin{array}{c} v_{G}\left(t\right)=v_{S}\left(t\right)+L_{F}\frac{\Delta I_{G}}{\Delta t} \end{array}$$
(3)$$\begin{array}{c} I_{G}=\frac{v_{G}\left(t\right)-v_{S}\left(t\right)}{L_{F}}\cdot \Delta t \end{array}$$

The control unit that switches the IGBT transistors inside the *CICS* modules is FPGA-based. As a result, the time required to generate individual samples acquired via voltage analog input is reduced. The current shape that is generated via the *CICS* module is the same as the shape of the voltage that is applied to the control analog input of the CICS module.

An *Intel Cyclone 10 LP* FPGA module is used in the *CICS* module. The FPGA calculates and regulates the current using a variable current hysteresis current control with constrained switching frequency algorithm. The frequency of current measurement and control loop operation is 1 MHz. Another task of the unit is to generate switching PWM pulses for the IGBT transistors. The switching frequency of the transistors is set to 50 kHz. The FPGA communicates with the master MCU unit via the SPI bus. The MODBUS RTU and TCP/IP communication interface is used for communication with other external peripherals (front panel, cDAQ system, PC).

The FPGA module also takes care of regulating the voltage level in the DC link, which is provided by the classic PID algorithm. The PID controller is set to PI with gain parameter Kp(-) = 0.2 and integration time constant Ti(s) = 0.045. The regulated system is astatic with the dominant capacitance of the intermediate circuit C = 1.5 mF and forms a second-order system with the regulator. For the basic setting of the system dynamics, a damping of 1.5 is considered, which was chosen as a compromise between the response speed and the behavior of the real system (the influence of traffic delays and other influences not included in the theoretical consideration, such as filters). The control frequency is 50 kHz. A simplified schematic diagram of the CICS module inner control system can be seen in Figure 4. A schematic diagram of the whole *CICS* module can be seen in Figure 5.

In the use-case scenario, the *CICS* system is considered by the user purely as a black box that generates the required current shape. There is no need for any additional settings.

### 2.2. Three-Phase Controllable Load

Load modules are basically the same as the aforementioned injection modules. Load modules inject defined current into the distribution network, which simulates with its shape of the waveform the real load. The main difference is that these modules can take the active power p(t) from the grid. This is ensured by connecting external resistors in which the active power is properly discharged.

In addition to load synthesis, it is also possible to connect a real appliance to the *CICS* system (up to 6 × 1-phase and 2 × 3-phase) that will be compensated.

### 2.3. Serial Inductors

Softening serial inductors can be connected to the distribution network to simulate negative voltage conditions. As a result, many negative voltage conditions scenarios can be simulated. From the point of view of filtering issues, the load is defined by the spectrum of the consumed current. Each harmonic component of the current causes a drop in the corresponding voltage component in the series inductance of the supply network. This voltage drop is proportional to the series inductivity and harmonic component frequency and affects the final load voltage. The measured series inductance of the electrical network connected to the experimental workspace was determined to be approximately 150 μH. Currently, the experimental workspace includes additional softening inductors, which can be connected in series with the power grid to increase the input inductance for observing the effects with different input inductance. The four values of additional installed inductors are 0 mH, 5 mH, 10 mH and 15 mH. A change in induction value can be performed manually by the user or remotely via a MODBUS command.

### 2.4. Measurement Acquisition

Data acquisition of required electrical quantities is carried out by the NI *cDAQ-9189* network chassis. A total of 12 currents (grid i(t), load $i_{L}\left(t\right)$, injection $i_{G}\left(t\right)$, neutral $i_{N}\left(t\right)$) and three voltages (grid v(t)) are measured and connected to the cDAQ acquisition system via signal conditioning modules (SCMs). The amplitude range of voltage modules is up to ±400 V and current modules is up to ±21 A. The purpose of the SCM is to provide signal conditioning to voltages in the ±10 V amplitude range that are correctly measured by the AI measurement modules over the full range. Four *NI-9215* AI modules are used for data acquisition, and two *NI-9263* AO modules are used for data generation to control the current generators and loads. The measurement system complies with the IEC 61000-4-30 standard (https://www.technicke-normy-csn.cz/csn-en-61000-4-30-333432-181262.html# (accessed on 24 July 2023)) used in the power quality field [40]. More information about adjustment and calibration of the data acquisition system is described in Section 3.

### 2.5. External Controller—Software

The power modules are controlled through an application that is programmed in the LabVIEW programming environment. The application itself runs on a higher-level system (PC) that communicates with the *CICS* platform via an Ethernet interface. Currently, the application can communicate with the power modules, perform measurements with respect to standards, carry out simple grid scenarios and implement various SAPF control methods. The application is designed to be modular due to the future extension of functions. The event-driven queued message handler (EQMH) was chosen as a suitable application architecture to meet all the requirements of the application. Details on this architecture are described in [41]. Communication between modules is maintained via first-in first-out (FIFO) messages. In order to make FIFO as usable and flexible as possible, every message/data point is transformed into the “variant” data type. One FIFO can be therefore used to send multiple data types, which is the main advantage of this approach. A simplified application diagram can be seen in Figure 6.

There are currently nine developed LabVIEW modules, which run in parallel and independently of each other.

#### 2.5.1. Graphical User Interface

This module maintains handling of every event created by the user (e.g., button click, keyboard stroke) and is considered as a main module. In its initialization state, the system calls and subsequently runs every other “slave” module. From its front panel, the user can control the whole application, access other subpanels, run scenarios, set the CICS parameters, turn output on/off or run the logger.

#### 2.5.2. MODBUS Communication Module

MODBUS module maintains communication with the machine which occurs in the background. Mainly, it converts the messages from the GUI module to MODBUS packets, which controls the *CICS* machine. The module can also restart the watchdog timer every 200 ms, or the timeout error occurs and the system enters an error state. This module in its default state is not accessible to the user and runs in the background.

#### 2.5.3. Continual Data Acquisition

The objective of this module is to continuously acquire all the available data at the DAQ buffer by using the DAQmx library functions. As stated in Section 2.4, there are total of 15 channels that must be read for each cycle run. Every channel is represented as a waveform of instantaneous data. Sampling frequency is set to 62.5 kS/s, and the time window $T_{W}$ is 200 ms. These data are then distributed to other processes for additional processing via FIFO. If an error occurs, the DAQ process restarts itself without restarting the main application. If the error persists, the main application is terminated.

#### 2.5.4. Calculations and Graphical Representation of Data

The main objective of this module is to inform the user about the current value of all electrical quantities that are measured or calculated. From sampled values of voltage and current frequency, U${}_{RMS}$ and I${}_{RMS}$, voltage and current spectra, group spectra, amplitudes and phases of individual harmonic components, apparent power, active power, reactive power and total harmonic distortion for voltage and current are calculated. All electrical data can be saved on local storage.

#### 2.5.5. Harmonic Current Extraction

This module takes care of application of harmonic current extraction (HCE) method on data. Every HCE method is considered as a separate entity, a function VI which is dynamically called on application startup. Due to this architecture, implementation of a new method into the application becomes easy. User can change the method on the fly. This process is precisely optimized so that the computational method is not burdened by any “unnecessary” operations. The processed current after the harmonic extraction is then adjusted to the actual phase of distribution network voltage ϕ(t) (if necessary) and then sent to other processes to be generated by the analog output module.

#### 2.5.6. Synchronization Module

This module functionality is partly optional because not every HCE method needs to “know” about actual voltage phase value. The main objective of this task is to correct the phase delay Δϕ caused by the measurement, calculation and injection process. Because the used HW is not ideal (cDAQ system controlled by PC) and the delay caused by the flow of information is not zero, the phase delay has to be compensated. It is necessary to implement a phase-locked loop (PLL) that locks to the phase value of the voltage signal [42].

Approximation delay of the system Δt can be calculated as:(4)$$\begin{array}{c} \Delta t=t_{\mathrm{ACQ}}+t_{\mathrm{HCE}}+t_{\mathrm{SYN}}+t_{\mathrm{GEN}}+t_{\mathrm{SHIFT}}\;\left(s\right), \end{array}$$
where $t_{\mathrm{ACQ}}$ is the time at which the data are acquired by analog input modules, buffered and sent to the PC via TCP-IP. $t_{\mathrm{HCE}}$ is the time at which the selected harmonic extraction method reaches its goal and calculates the reference current to be generated; the value of this delay is dependent on the method used. $t_{\mathrm{SYN}}$ is the time until the PLL Δϕ reaches zero value. $t_{\mathrm{GEN}}$ is the time needed for all samples to be generated by the analog output module. Finally, $t_{\mathrm{SHIFT}}$ is the delay caused by the phase shift of the waveform. This shift occurs because the ratio between the sampling frequency $f_{s}$ and frequency of the grid *f* is not an integer. The waveform then tends to be “sliding” inside the measurement window.

#### 2.5.7. Continual Generation of Injection Current

After all calculations and corrections of the injection signals are carried out, they are finally generated by the analog output module. Because the generation is not executed point by point, it is necessary to truncate this process to a minimum. The goal of this process is to generate the required samples of the injection reference signals that are sent via FIFO from other process. Other necessary operations are carried out outside this task because the DAQ output operations are very sensitive to buffer overflow. The sampling frequency $f_{S}$ is 62.5 kS/s, but the size of the buffer is set only for 5 kS. If data are not available in time, the timeout state is initiated, which is considered by the application as a error, thus terminating the process.

#### 2.5.8. Controllable Load Control

This part of the application takes care of execution of scenarios. A scenario is a set of *CICS* system parameters that can change over time. The parameters are configured by the definition text file in the form of a table. The format of the input table can be seen in Table 1. The text table is processed and converted to a series of generated waveforms. This software module keeps track of the elapsed time and sets the value of individual phases according to the current state of the scenario through the analog output voltage. Each phase can be set independently of others. Every harmonic up to the 50th can be set in the form of a percentage value from the fundamental amplitude of the current.

#### 2.5.9. Continual Data Logger

This module continuously saves the acquired data into the NI *TDMS* data file structure. Channels are stored in the form of the waveforms and are divided into voltage and current groups. All calculated data (from Section 2.5.4) are included in the file.

## 3. CICS Measurement Part Accuracy and Calibration

In terms of measurement, it is necessary to mention the accuracy of the measuring system. As mentioned in Section 2.4, signal measurement and control voltage generation are provided by the NI *cDAQ-9189* chassis containing *NI-9215* analog inputs and *NI-9263* analog outputs. The *NI-9215* contains four differential analog inputs, 16-bit A/D converters, 100 kS/s sampling rate, simultaneous sampling and ±10 V range. *NI-9263* includes four differential analog outputs, 16-bit DACs, 100 kS/s sampling rate and ±10 V range. NI, as the manufacturer of the analogue modules *NI-9215* and *NI-9263*, provides their error limits, which can be seen in Table 2.

The measured signals are adjusted by the signal conditioning module (SCM) before being connected to the *NI-9215* ADCs. The SCM is part of the CICS platform and its characteristics are determined by the circuit design proposed by the CICS manufacturer. The electronics of the SCM are not ideal from a metrological point of view and introduce amplitude errors, phase errors and frequency dependence into the measurement chain, mainly caused by the integrated anti-aliasing filter. The IEC 61000-4-30 Power Quality Analyzer Measurement Accuracy Standard was used to determine the limiting error of the measurement chain. In order to meet the required accuracy of the measurement chain (SCM + AD converter), an automatic adjustment must be performed to detect non-linearities, phase shifts and frequency dependence. A special software has been developed for the adjustment in the LabVIEW environment, and the measurement chain can be seen in Figure 7.

The resulting calibration report is used to verify the accuracy and to determine whether the detected errors fall within the error limits of the IEC 61000-4-30 standard. The constants detected during the adjustment are then applied in the measurement software, which processes the data from the AD converters to minimize the errors detected by the adjustment. To verify the accuracy of the entire measuring system, including the measuring software, calibration was performed after adjustment. The calibration provided information about errors of the measurement chain used for CICS control purposes and resulted in an accuracy verification report that the measurement system met the requirements of the measurement and test technology standard 61000-4-30. The results of the adjustment and calibration cannot be included in this article due to the large quantity of data (calibration of voltages and currents in five parts of the range and also in the frequency range (2nd–50th harmonic) on each part of the range). The OMICRON CMC 256plus calibrator, which was used for both the adjustment and calibration, is capable of generating a voltage of 0–300 $V_{rms}$ and a current of 0–12.5 $A_{rms}$. For this purpose, this calibrator is sufficient in terms of accuracy and also range. The accuracy of this calibrator is shown in Table 3.

## 4. Implemented Methods

In order to conduct a verification of the *CICS* system, the Notch–least mean squares (Notch–LMS) and Notch–recursive least squares (Notch–RLS) algorithm were chosen as candidates for verification. Compared to other classical time-domain methods, these algorithms are difficult to implement, computationally demanding and require proper setup for optimal functionality. An adaptive filter with this structure is able to process multiple reference input signals $x_{i}\left(n\right)$ from different sources. Every reference input has its own weight coefficient $w_{i}$ [27].

Notch filters were chosen, since one of the parameters (filter order) is hardcoded. The implementation is also easier (the memory requirements are also lower). Both filters are adaptive and can therefore react to any changes to noise characteristics (usually sooner than one period of signal).

### 4.1. Notch–Least Mean Squares

Notch–LMS is a modification of the original LMS [43] algorithm. There are three basic steps in this algorithm:(5)$$\begin{array}{c} y\left(n\right)={\vec{w}}_{1}^{T}\left(n\right)\cdot {\vec{x}}_{1}\left(n\right)-{\vec{w}}_{2}^{T}\left(n\right)\cdot {\vec{x}}_{2}\left(n\right), \end{array}$$(6)$$\begin{array}{c} e(n)=d(n)-y(n), \end{array}$$(7)$$\begin{array}{c} \begin{array}{c} {\vec{w}}_{1}\left(n+1\right)={\vec{w}}_{1}\left(n\right)+2\mu \cdot e\left(n\right)\cdot {\vec{x}}_{1}\left(n\right),\\ {\vec{w}}_{2}\left(n+1\right)={\vec{w}}_{2}\left(n\right)+2\mu \cdot e\left(n\right)\cdot {\vec{x}}_{2}\left(n\right), \end{array} \end{array}$$
where y(n) is the output of FIR filter, e(n) is the estimated error signal, ${\vec{w}}_{i}\left(n\right)$ are the weight coefficients, μ is the convergence constant (step size) and $x_{i}\left(n\right)$ are the reference input signals.

It is worth mentioning that the choice of the μ significantly influences the function of the filter. A larger value of the μ results in loss of stability and also inaccurate filtration. On the other hand, lower values of the convergence constant cause an increase in computation time and augment computation demands.

With lower values of μ, the system that uses the LMS algorithm is able to achieve good results in terms of lowering the THD${}_{I}$ value, but the convergence time takes too long. On the other hand, the higher the μ gets, the more dynamical the system becomes, but it loses its stability in terms of mitigating higher harmonic components, thus being ineffective.

The input signal *d(n)* to the LMS algorithm is current phase, and *x(n)* and *x(n)${}_{90^{\circ }}$* are reference signals. In order to generate reference signals, it is necessary to perform Clarke transformation, which transforms three-phase currents from *abc* to αβ coordinates:(8)$$\begin{array}{c} \left[\begin{array}{c} i_{\alpha }\left(t\right)\\ i_{\beta }\left(t\right)\\ i_{0}\left(t\right) \end{array}\right]=\sqrt{\frac{2}{3}}\cdot \left[\begin{array}{ccc} 1 & -\frac{1}{2} & -\frac{1}{2}\\ 0 & \frac{\sqrt{3}}{2} & -\frac{\sqrt{3}}{2}\\ \frac{1}{\sqrt{2}} & \frac{1}{\sqrt{2}} & \frac{1}{\sqrt{2}} \end{array}\right]\cdot \left[\begin{array}{c} i_{a}\left(t\right)\\ i_{b}\left(t\right)\\ i_{c}\left(t\right) \end{array}\right]. \end{array}$$

After the transformation, it is necessary to filter out $i_{\alpha }\left(t\right)$ and $i_{\beta }\left(t\right)$ from the fundamental harmonic component. These signals are then considered as *x(n)* and *x(n)${}_{90^{\circ }}$*.

After execution of the Notch–LMS algorithm, the negative value of error signal e(n) is then considered as $i_{ref}\left(n\right)$ and is generated via the analog output module.

### 4.2. Notch–Recursive Least Squares

As with the Notch–LMS algorithm, the Notch–RLS algorithm is based on the original RLS algorithm [43]. The output of the FIR filter y(n) equation is similar to Equation (Equation 5), but the weights are recursively calculated from past iterations. Mean gain vectors $k_{1}\left(n\right)$ and $k_{2}\left(n\right)$ are: Response “We confirm this is correct.”
(9)$$\begin{array}{c} \begin{array}{c} {\vec{k}}_{1}\left(n+1\right)=\frac{P_{1}\left(n\right)\cdot {\vec{x}}_{1}\left(n+1\right)}{\lambda +{\vec{x}}_{1}^{T}\left(n+1\right)\cdot P_{1}\left(n\right)\cdot {\vec{x}}_{1}\left(n+1\right)},\\ {\vec{k}}_{2}\left(n+1\right)=\frac{P_{2}\left(n\right)\cdot {\vec{x}}_{2}\left(n+1\right)}{\lambda +{\vec{x}}_{2}^{T}\left(n+1\right)\cdot P_{2}\left(n\right)\cdot {\vec{x}}_{2}\left(n+1\right)}. \end{array} \end{array}$$

Estimated error value e(n) is:(10)$$\begin{array}{c} e(n)=d(n)-y(n). \end{array}$$

Weights $w_{i}\left(n\right)$ of FIR filter are updated according to:(11)$$\begin{array}{c} \begin{array}{c} {\vec{w}}_{1}\left(n+1\right)={\vec{w}}_{1}\left(n\right)+e\left(n+1\right)\cdot k_{1}\left(n+1\right),\\ {\vec{w}}_{2}\left(n+1\right)={\vec{w}}_{2}\left(n\right)+e\left(n+1\right)\cdot k_{2}\left(n+1\right). \end{array} \end{array}$$

Inverse matrices $P_{1}\left(n\right)$ and $P_{2}\left(n\right)$ are calculated with:(12)$$\begin{array}{c} \begin{array}{c} P_{1}\left(n+1\right)=\lambda^{-1}P_{1}\left(n\right)-\lambda^{-1}{\vec{k}}_{1}\left(n+1\right){\vec{x}}_{1}^{T}\left(n+1\right)P_{1}\left(n\right)\\ P_{2}\left(n+1\right)=\lambda^{-1}P_{2}\left(n\right)-\lambda^{-1}{\vec{k}}_{2}\left(n+1\right){\vec{x}}_{2}^{T}\left(n+1\right)P_{2}\left(n\right) \end{array} \end{array}$$

The functionality of this algorithm is fundamentally influenced by the correct choice of the forgetting factor λ value. The forgetting factor is in the range λ∈(0,1]. If the forgetting factor equals 1, then the estimation is performed without forgetting. In practical implementation, we usually consider range of values from λ=0.98 to λ=1.00. The value of λ coefficient is directly related to the contribution of previous samples to the covariance matrix. This results in the filter being insensitive to the past samples.

## 5. Experimental Verification

In order to prove the functionality of the whole *CICS* system, a series of experiments were conducted. Figure 8 shows 30 s of $I_{L}\left(t\right)$ current with which all experiments were performed in order to investigate the behavior of the harmonic extraction methods with step changes in the current drawn by the loads. Important transients are marked there. Transient *A* shows the switching of the load from standby mode to run mode. *B* shows change from a lower to higher load current value. Next, *C* shows the transient of switching inventor from the run mode back to the standby mode. Finally, the *D* shows the same transient as *A* but for a much higher current value. In standby mode, the presence of distortion is detected due to current draw by the electronics to power the control electronics and network elements. The amplitude of the fundamental harmonic component of the current $I_{L}\left(t\right)$ in standby mode is 0.1 A. Current distortion is caused by the 7th and 15th harmonic components of the current with an amplitude of 10% of the fundamental harmonic component and unwanted noise. All experiments on the CICS platform were performed in three phases, but, for better description and representation in the graphs, data from only one phase L1 are shown.

Because of the nature of the *CICS* system application, the main criterion according to which all experiments were evaluated was the average relative THD${}_{I}$ improvement. It is calculated as the ratio between the reference (original) THD${}_{I}$ value of one cycle and its compensated counterpart, expressed as a percentage.

Total current harmonic distortion is calculated as the ratio of the sum of all higher current harmonics and current fundamental harmonic value $I_{1}$. This is all according to:(13)$${\mathrm{THD}}_{I}=\frac{\sqrt{I_{2}^{2}+I_{3}^{2}+I_{4}^{2}+\ldots +I_{N}^{2}}}{I_{1}^{2}},$$
where $I_{1}$ is the value of the fundamental harmonic component. The currents with an index higher than 1 are higher harmonic components. *N* is the highest considered harmonic component.

Due to transients, the THD${}_{I}$ value cannot be calculated from the entire signal. The signal must be separated into individual periods. Individual THD${}_{I}$ values are then calculated from these periods. This process is performed both for the current signal before compensation and for the one after compensation is performed. This results in an original non-compensated ${\mathrm{THD}}_{REF}$ value and its compensated counterpart ${\mathrm{THD}}_{COMP}$ in the form of vectors.

From these vectors, relative improvement (marked as δ) of THD${}_{I}$ value in percentage is calculated according to:(14)$$\begin{array}{c} \delta_{j}=\frac{{\mathrm{THD}}_{REF_{j}}-{\mathrm{THD}}_{COMP_{j}}}{{\mathrm{THD}}_{REF_{j}}}\cdot 100\left(\%\right), \end{array}$$
where *j* is the index of the investigated current signal period. This value is then used as a criterion that assesses the quality of SAPF filtering in a given period.

Furthermore, as the last step, the average relative THD${}_{I}$ improvement value ø δ THD${}_{I}$ is calculated as:(15)$$\begin{array}{c} ø\;\delta \;{\mathrm{THD}}_{i}=\frac{1}{N}\sum_{j=1}^{N}\delta_{j}\;\left(\%\right), \end{array}$$
where *N* is the quantity of the all periods contained in the examined current signals. This result can be used to assess the overall average active filtration result in the experiment.

### 5.1. *CICS* System Verification

In order to prove system functionality, namely the higher harmonic current component mitigation and thus the reduction in THD${}_{I}$, a basic experiment with signal generation was performed. As the *CICS* system allows the synthesis of the load through the controllable analog input, the reference signal for calculation of THD${}_{I}$ reduction can be easily generated. The “method” for this experiment is presented as “ideal”, since both the load current signal and its injection counterpart, without the fundamental harmonic, are predetermined in advance. The results of this simple experiment can be seen in Figure 9. In this case, the reference values of THD${}_{I}$ contain only phase-locked loop time shift error because of the non ideal state of the used system (cDAQ controlled by the PC) and because the used sampling frequency $f_{S}$ is not an integer multiple of the grid frequency *f*. This causes continual shifting of the measured waveform that needs to be compensated. This error does not contain any delay originating from HCE method calculation or SW because the injection current waveform is known and prepared in advance.

To make the experiment more meaningful, it is necessary to simulate the HCE method calculation delay. To simulate this, the delay of $2T_{W}$ (two lengths of measurement window) has been incorporated. When applying the SAPF system, the most critical point is the transient between two steady states. The error is influenced by the delay of the phase-locked loop, as well as the HCE calculation method itself which is used to adapt to the new situation. However, this causes additional detuning of the phase-locked loop. Results with this incorporated delay can be seen in Figure 10. The reduction in the THD${}_{I}$ has been little degraded, approximately by 2%.

### 5.2. Notch–LMS Algorithm Verification

As stated above, the adaptive algorithms from the LMS branch are very sensitive to the value of the convergence constant μ, which significantly affects its functionality. Because of this, the dependency between μ and the resulting THD${}_{I}$ was examined. All settings of the μ and its result on the average relative THD${}_{I}$ improvement can be seen in Table 4.

For this experiment, the optimal value of the convergence constant is μ = $1\cdot 10^{-6}$. It should be noted that the optimal settings will be different for each experiment, but it is possible to specify a range of values where the adaptive algorithm will work properly. Furthermore, the value of the μ parameter can be tweaked on the fly, but this would need an additional layer of the control above the method itself.

Figure 11 shows the THD${}_{I}$ values after the application of the Notch–LMS algorithm. Significant improvement in reduction in the THD${}_{I}$ can be seen. However, when the transient occurs, the reduction decreases due to the fact that the phase-locked loop is detuned and the system tries to adapt to a new state. When the state is steady (the PLL is locked), the THD${}_{I}$ is improved and decreases to the values below 5%.

Figure 12 shows the Notch–LMS application and its behavior in transient *A* (taken from Figure 8). The waveforms show the transition between standby and run mode. It is worth mentioning that due to the architecture of the application, there is a delay of two time windows and additional delays due to the communication, generation and other factors. Compensation starts at 0.48 s, but the filtering starts to work properly around 1.4 s. Adaptation time is therefore approximately 900 ms.

Figure 13 shows the Notch–LMS application and its behavior in transient *B* (taken from Figure 8). The plot shows the transition from the lower current value to the higher one with different harmonic settings. Again, the delayed compensation is noticeable, but this time there is a partial compensation included. This is due to the fact that after the load change in the time of 5.55 s, the compensation works for previous load settings up to the time of 5.6 s. Then, the compensation takes the correct form, but it takes approximately 500 ms before filter enters the steady state.

Figure 14 shows the Notch–LMS application and its behavior in transient *C* (taken from Figure 8). The waveforms show the transition from run mode to standby mode. Due to the latency, there is an unwanted injection into the grid which lasts for approximately 225 ms. This causes the SAPF to decrease power quality instead of improving it, which is undesirable.

Figure 15 shows the Notch–LMS application and its behavior in transient *D* (taken from Figure 8). In contrast to transient *A*, the phase-locking in this case takes longer to enter the steady state. In this case, the correct compensation does not take effect until approximately 2 seconds later. This delay is higher than in the case of transient *A* by 1.1 s.

### 5.3. Notch–RLS Algorithm Verification

Adaptive algorithms from the RLS family require correct setting of the forgetting factor for their optimal function. In case of power quality applications, the value of the forgetting factor tends to be close to the value of 1. Examined values in the experiment were in the range from 0.9 to 1. All values set and their corresponding results can be seen in Table 5. For this experiment, the optimal value of the forgetting factor is 0.9999.

Figure 16 shows the THD${}_{I}$ values before and after the Notch–RLS application. As with the Notch–LMS algorithm, a significant reduction in the THD${}_{I}$ can be noticed. The ability to mitigate THD${}_{I}$ is reduced during transients. The RLS algorithm shows slightly better results than the LMS algorithm.

As for individual transients, the behavior of the Notch–RLS algorithm is very similar to the Notch–LMS. In transient *A*, RLS algorithm shows better results in adaptation time. The algorithm reaches the steady state 200 ms faster. Transients *B* and *C* shows same results as LMS. Convergence time is approximately the same. Transient *D* is processed faster by the RLS algorithm than LMS, again approximately by 200 ms.

## 6. Discussion

The quality of the resulting compensation is influenced by a number of factors—i.e., the type of the used adaptive algorithm, the setting of the phase delay or the latency of the filter control unit (FCU). Experiments have shown that the currently implemented filter control unit does not meet the expected functionality of a compensator of higher harmonic components under all operating conditions. The reasons for the imperfect functionality of *CICS* are twofold: (1) the non-RTOS used in the FCU and (2) the HW cDAQ used to measure and generate signals. Thus, the FCU exhibits significant latency. *CICS* shows good results of suppressing harmonic components in the steady state of the distribution network with compensation. However, during transients, compensation is experienced with a delay of up to 2 s, which corresponds to approximately 100 periods of the distribution network signal in a 50 Hz system. This delay is unacceptable.

To improve the *CICS* functionality, it is necessary to replace the non-RTOS FCU based on a Windows PC and cDAQ with other technology that has a lower latency. For comparison purposes, experimental latency measurements of three types of FCUs based on NI hardware components were performed. The following variants were compared: (1) Windows PC and NI PCI-6221, (2) Windows PC and NI cDAQ-9185 Ethernet chassis, (3) NI cRIO-9039 RTOS controller with the FPGA.

### 6.1. Latency Comparison of Three Control System Variants

The scheme of the comparison experiment can be seen in Figure 17. The function generator generates a rectangular signal $v_{1}\left(t\right)$ with increasing amplitude. This signal is digitized by the analog input of the FCU into a sample vector $v_{1}\left(n\right)$. The sample vector $v_{1}\left(n\right)$ is immediately generated by the analog output of the FCU without any modification and takes the form of a voltage $v_{2}\left(t\right)$. Signals $v_{1}\left(t\right)$ and $v_{2}\left(t\right)$ are connected to an oscilloscope where the delay Δt is measured between individual signals.

#### 6.1.1. Windows PC and NI PCI-6221

NI *PCI-6221* is a multifunction DAQ board that has 16x AI and 2x AO. The resolution of both the ADC and DAC is 16 bits. The sampling rate is up to 250 kS/s. To minimize FCU latency, the DAQ card’s HW-timed single-point mode of operation has been used for both analog input and analog output.

In this mode, samples are measured and generated continuously sample-by-sample using hardware timing and without any buffers. Due to the lack of a dedicated buffer, it is necessary to ensure that reads and writes are only performed at the speed that is suitable for hardware timing of a Windows PC with non-RTOS. If an excessive sampling rate is requested or if the filter algorithm takes too long to calculate the value of the generated sample, a DAQ process error will be raised.

The programmer must ensure that the computation time per sample of the filter algorithm is lower than the inverse of the desired sampling rate $\frac{1}{f_{S}}$. The computation of the filter algorithm must be sufficiently fast to keep up with the HW timing. Since the computation time per sample of the filter algorithm depends on the chosen algorithm (method), the maximum sampling rate $f_{S}$ is also directly connected to the used method.

For comparison purposes, the sampling rate $f_{S}$ was gradually increased throughout the experiment, while the FCU latency was measured. The FCU was not calculating any harmonic extraction algorithms, and the samples were immediately generated at the analog output. The results are shown in Table 6.

Based on the results, it is clear that the HW timing in HW-timed single-point mode is not accurate and is influenced by jitter with a variance of units to tens of μs. The DAQmx library offers a partial solution, as there is a function that can be used to eliminate jitter—*Wait For Next Sample Clock*. However, after implementing this function, it was not possible to set the value of the sampling rate $f_{S}$ above 1 kHz—this setting is, however, unsuitable for SAPF purposes. In the case of the HCE method implementation, the jitter was additionally increased by the variable delay caused by the computation of the implemented method.

#### 6.1.2. Windows PC and NI cDAQ-9185 Ethernet Chassis

CompactDAQ is a modular measurement and signal generation platform consisting of a chassis and optional I/O modules. cDAQ-9185 is a four-slot chassis with a 2x Gb Ethernet interface. Timing and synchronization is provided by an internal clock. The NI-9215 module (4x simultaneous analog inputs, 16-bit ADC, 100 kS/s) and the NI-9263 module (4x simultaneous analog output, 16-bit DAC, 100 kS/s) were used to detect FCU delay. These identical modules are part of the FCU in the CICS platform for measurement and signal generation.

The cDAQ communicates with the Windows PC via TCP/IP protocol and uses *N*-sample methods based on a dedicated buffer. The time required to perform one iteration of data acquisition and subsequent generation includes the following subparts:$t_{1}$: Measuring the data via the AD converter and inserting the data into the internal input buffer of the cDAQ system.$t_{2}$: Sending the buffered data via Ethernet to the input buffer of the RAM memory in the PC.$t_{3}$: Calculation of harmonic extraction algorithm and buffering of data into the output buffer of RAM memory in the PC.$t_{4}$: Sending buffered data via Ethernet to the internal output buffer of the cDAQ system.$t_{5}$: Generating data available in the cDAQ output buffer using the DAC module.

For the visual representation, refer to Figure 18.

The measurement window length $T_{W}$ is based on the ratio of the readings *N* and the sampling frequency $f_{S}$. The algorithm calculation time between each iteration of the control loop cannot exceed the $T_{W}$, since the buffer would accumulate unread samples.

The analog output operation is based on a different principle than the analog input operation. The program writes data to the buffer and must deliver the new data before the buffer is emptied. If this does not happen, the data are lost. It follows that the sum of $t_{4}+t_{5}$ must be smaller than the sum of $t_{1}+t_{2}+t_{3}$, which is difficult to achieve. Times $t_{2}$ and $t_{3}$ are directly proportional to the length of the measurement window $T_{W}$.

One possible solution is to minimize the length of $T_{W}$ by setting a low number of *N* sample readings. The combination of a high sampling rate $f_{S}$ and a small number of *N* samples is not realistic, since the speed of frequent communication of small quantities of data over Ethernet is not sufficient. Another solution would be to reduce the value of the sampling rate $f_{S}$ and leaving the number of *N* samples at a higher value. In this case, the frequency band of the measured signal would be reduced, and thus the quality of the compensation would be degraded. It is also possible to increase the size of the PC’s output RAM buffer. This will in turn increase the latency, which is also unsuitable.

One of the advantages of this approach is that the implementation in a LabVIEW programming environment is very simple, mainly thanks to the DAQmx library. However, this approach has its own significant disadvantage—the amount of latency is at least the sum of the length of the measured window $T_{W}$ and the communication latency. A comparison of the different settings of this solution can be seen in Table 7 and Table 8. Harmonic current extraction cannot be performed on this approach due to the unacceptably high latency, even though efforts were made to achieve a compromise between the sampling rate and the length of the measurement window $T_{W}$.

#### 6.1.3. NI cRIO-9039 RTOS Controller with FPGA

The CompactRIO (cRIO) platform consists of a controller with RTOS, an FPGA chassis and individual I/O modules. The cRIO controller is a high-performance controller that deterministically executes LabVIEW code in real time (RT). The platform chassis is directly interfaced with I/O to achieve high performance. The first two modules are directly connected to the FPGA chip and do not communicate through the bus. This design achieves almost negligible system response latency compared to the previously described architectures. The rest of the modules communicate through the SPI bus.

The *cRIO-9039* contains a 1.91 GHz Intel Atom processor with four cores and an NI Linux RTOS operating system. One of the main parts of the cRIO platform is a 40 MHz Xilinx Kintex FPGA. The measurement modules used are the same as in the previous experiment. The principle for RTOS processors with the FPGA is different from the cDAQ system because the code can run on both the processor and the FPGA. There is no operating system on the FPGA itself, and the programmer does not have to deal with the communication latency over the device bus. The code for the FPGA takes the form of a logic circuit that is implemented directly on the hardware layer ensuring that the system latency can be considered constant. The LabVIEW programming environment allows the programmer to create code for the FPGA, but the code must be translated from LabVIEW to HDL and then synthesized using Xilinx. The disadvantage of developing code for FPGAs is the difficulty of programming and debugging and also the increase in compile time of the entire program as the code is extended.

Another significant disadvantage of FPGA code is the lack of floating-point data type operations. All numeric operations must be converted to a fixed-point data type (FXP). The conversion of these operations must be performed manually, and if errors occur in the code it can lead to significant inconsistencies of the achieved results.

Despite these major disadvantages, this approach is inevitable when developing and working with the RT controller *cRIO-9039* containing FPGAs. The controller itself was verified in a comparison experiment, where the latency reached 12 μs. This latency was caused by the used I/O modules, which can only operate at a maximum sampling rate of 100 kS/s. The inverse of this rate corresponds to the minimum latency of 10 μs. This latency is more than sufficient for the purpose of the SAPF compensator.

### 6.2. Result of Comparison

The presented comparison showed that out of the three HW platforms, only the cRIO-9039 controller can meet the high demands required for SAPF compensation. Solutions that include a Windows PC as a control system are not applicable mainly due to the non-deterministic nature of the operation system that has to buffer data, which directly increases latency of the whole platform. The *cDAQ-9189* Ethernet Chassis controller currently installed in the CICS platform will remain for voltage and current measurements and subsequent analysis and evaluation of experiments. The CICS platform will be extended with the cRIO-9039 controller to measure voltage and harmonically biased current in the network, perform harmonic extraction according to the implemented compensation method and then generate a reference current with a latency in units of microseconds.

### 6.3. Active Filter Implementation on NI cRIO-9039

To verify the earlier statements, an implementation of the active filter on the *cRIO-9039* platform was performed. A three-phase distribution network containing harmonic voltage components was simulated using two NI-9263 modules with analog outputs. The FCU of the compensator consists of a *cRIO-9039* equipped with an *NI-9215* analog input module and an NI9263 analog output module. The Notch–LMS compensation method was chosen. A schematic diagram of the experimental setup can be seen in Figure 19, while the photograph can be seen in Figure 20.

The results from the experiment can be seen in Figure 21. Figure 21a shows the current distorted by higher harmonic components. Figure 21b shows the injection current generated by the *cRIO-9039*. Figure 21c shows the current signal after mathematical compensation. Mathematical compensation was carried out by summing the signals $i_{S}\left(t\right)+i_{G}\left(t\right)$ from the measured data.

The quality of the compensation was evaluated based on the resulting THD${}_{I}$. Results in Table 9 clearly show that cRIO-9039 is suitable for use as an FCU for SAPF functionality. The value of THD${}_{I}$ was on average improved by 97.75%.

### 6.4. Efficiency Evaluation of Notch Family Algorithms

A great influence on the functionality of the used adaptive algorithms Notch–LMS and Notch–RLS depends on the correct setting of their input parameters. As far as Notch–LMS is concerned, this algorithm requires a correct value of the convergence constant μ. For Notch–RLS, the most important parameter is the forgetting factor λ. The filter order for both of these algorithms is fixed to 2. With the correct setting of the aforementioned parameters, the minimum value of error is achieved. The results of the effect of setting the mentioned parameters can be seen in Table 4 and Table 5.

However, the problem is that it may be necessary to set the parameters differently for different tasks or signals. Therefore, it can be concluded that there is no universal value for these parameters, and the user is usually forced to set them empirically according to experience. Another solution that is offered is the use of optimization techniques that would automatically set the aforementioned parameters and therefore make the system work without the added error of the human factor. It should be emphasized that most optimization techniques are computationally intensive, but, due to the implementation of the algorithms on the FPGA, the speed of computation is greatly accelerated.

### 6.5. Further Research

After augmenting the *CICS* platform with the *cRIO-9039* controller, the next area of research is a more detailed evaluation of experimental measurements such as convergence time, algorithm stability, computational complexity, response to different types of transients and ability to operate under different distribution network conditions.

The main goal of the ongoing research is the implementation of additional types of methods, modifications of individual methods for given load conditions, and combination of methods, which should provide better results under certain conditions of real distribution networks. The following approach was chosen to further increase the number of sites for potential SAPF deployment on real distribution network cases and thus move the research on SAPF issues forward:1.Comprehensive measurements using a PQ mobile analyzer in production plants.2.Power load analysis of specific production machines.3.Scenario simulation on SAPF programmable loads.4.Deployment of methods on a specific case.5.Optimization and transfer back to the production plant.

With sufficient experimental measurements and testing using various harmonic extraction methods on different states of the distribution network, useful and applicable results can be obtained. These results can lead to the extension and deployment of SAPFs to manufacturing plants or public and private deployments in major facilities, such as production halls, GIGA factories, public lighting, shopping malls, hospitals, server rooms, schools, office buildings and supermarkets.

## 7. Conclusions

This paper presents a uniquely designed and constructed experimental platform, *CICS*, which allows testing and validation of harmonic extraction methods. With its open architecture and unique design (controllable current generators, controllable synthetic loads, SCM electronics, virtual instrumentation-based control system, etc.), the *CICS* platform can be used to evaluate the quality of SAPF control algorithms in different scenarios of power grid parameters while maintaining a uniform background. The main advantage of the *CICS* platform is the use of controllable synthetic loads that ensure repeatability of measurements and experiments, as well as the possibility to set up different scenarios of the power grid. This also allows for further and detailed research through testing and conducting experiments of both control and synchronization harmonic extraction methods. Prior to the actual experimentation on the *CICS* platform, adjustment and calibration were performed to ensure the adjustment of the calibration constants, which were then used in the software for the actual measurement by the platform controller. The subsequent calibration verified that the measurement system meets the measurement conditions of IEC 61000-4-30.

Subsequently, verification of the capability of the designed *CICS* system to correctly filter the harmonic components of the current and thus reduce the value of the total harmonic distortion in the current signals flowing through the network was performed. For this verification, the Notch–LMS and Notch–RLS adaptive algorithms were selected and used. In order to make the verification meaningful, the experimental verification of both algorithms was performed on step changes in load parameters (step changes in current draw) and also switching the load from standby mode to run mode and back.

The main and only criterion for successful classification was the average relative improvement in THD${}_{I}$. The Notch–LMS algorithm was able to reduce the THD value by 22.073% when the convergence constant μ was set to $1\cdot 10^{-6}$. The Notch–RLS algorithm provided slightly better results with a THD improvement of 22.346% for which the forgetting factor was set to 0.9999. Due to the nature of the LMS/RLS algorithms used (Notch family), the filter length is set to 2. Higher values have always had a negative effect on system performance.

Despite the very similar results of both methods, the Notch–LMS algorithm is currently the main candidate for real deployment. Compared to the Notch–RLS algorithm, it is simpler to implement, has a lower computational complexity and only one parameter needs to be set. It is also conventionally known and often used in other areas of signal filtering. The disadvantage of the Notch–RLS method is its high sensitivity to the correct setting of the forgetting coefficient λ. Despite the fact that both methods show similar values of relative THDI improvement in harmonic extraction results, the above facts about each method make it preferable to implement the Notch–LMS algorithm.

The quality of the resulting filtering is influenced, for example, by the type of adaptive algorithm used, the setting of the phase delay or the delay of the filter controller. According to the results of the experimental verification performed, it was found that the currently implemented NI *cDAQ-9189* filter controller does not meet the expected functionality of the higher harmonic components of the current under all operating conditions. The conclusion of the experiment is the necessity of using a filter controller with RTOS, which has lower latency, for the filtering of harmonic components and correct functionality of the whole *CICS* system. Hence, another experiment was conducted to verify and compare the latency for three types of filtering controllers based on NI hardware components.

Throughout the experiment, it was discovered that the filtering controller showing the lowest latency between signal measurement using AI and signal generation using AO is the NI *cRIO-9039* RTOS controller with an FPGA that had latency of 12 us. Finally, it was verified that the latency between signal measurement, applying the filtering algorithm and signal generation, is sufficient. From the results, it was verified that the cRIO-9039 is suitable for use as an FCU for the SAPF function in the *CICS* experimental platform. The THD${}_{I}$ value was improved by 97.75% on average. 

## Figures and Tables

**Figure 1 sensors-23-08494-f001:**
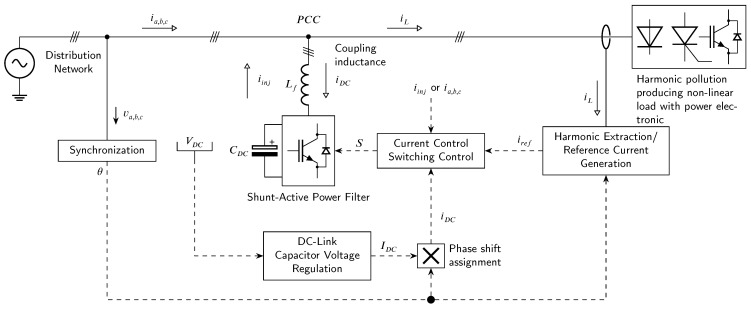
Simplified block diagram and working principle of SAPF [13].

**Figure 2 sensors-23-08494-f002:**
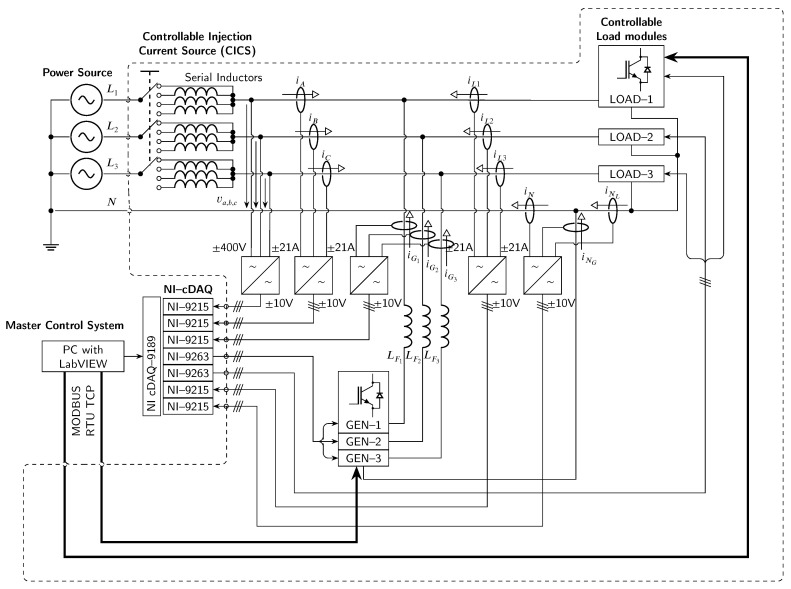
Controllable injection current source experimental platform schematic diagram.

**Figure 3 sensors-23-08494-f003:**
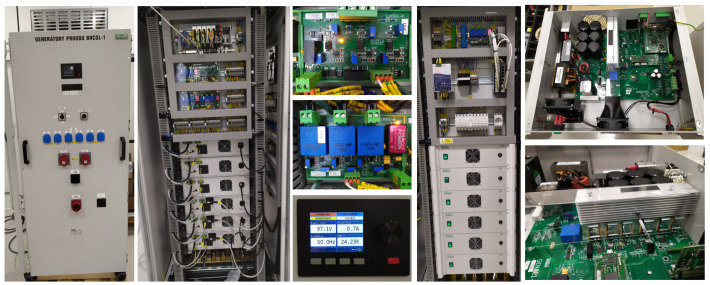
*CICS* platform photo collage.

**Figure 4 sensors-23-08494-f004:**
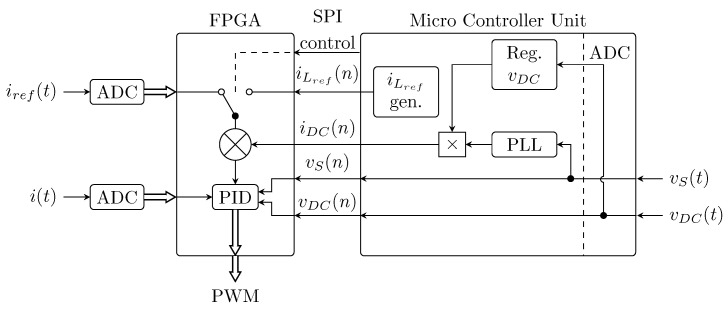
Simplified schematic diagram of inner controller (FPGA + MCU) functionality.

**Figure 5 sensors-23-08494-f005:**
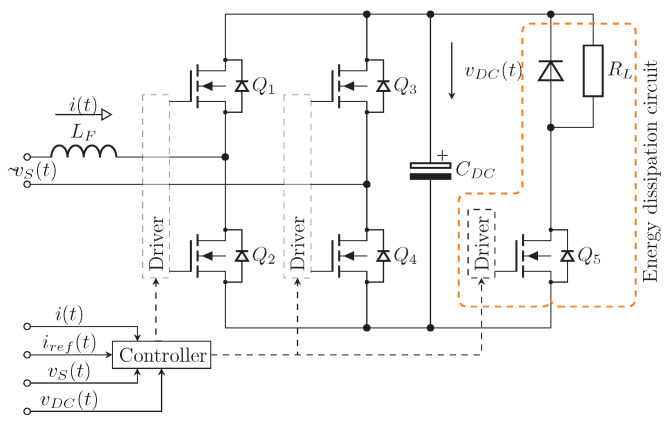
Simplified schematic diagram of the *CICS* module. Energy dissipation circuit is included only in load modules (orange polygon).

**Figure 6 sensors-23-08494-f006:**
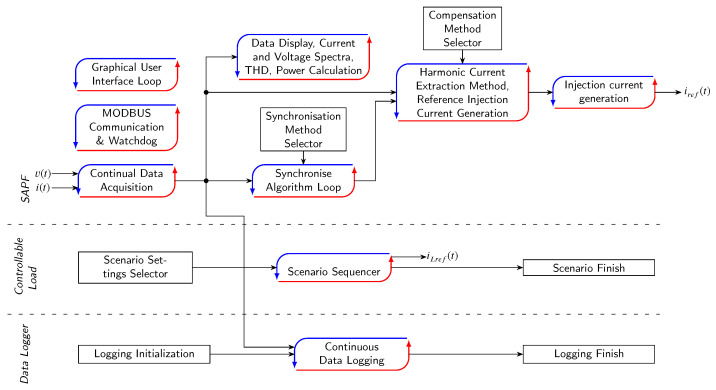
Simplified application flow diagram.

**Figure 7 sensors-23-08494-f007:**
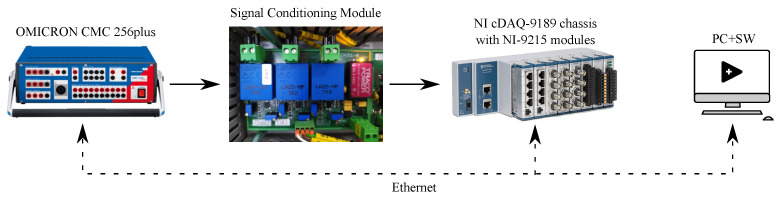
Measurement chain during adjustment and calibration.

**Figure 8 sensors-23-08494-f008:**
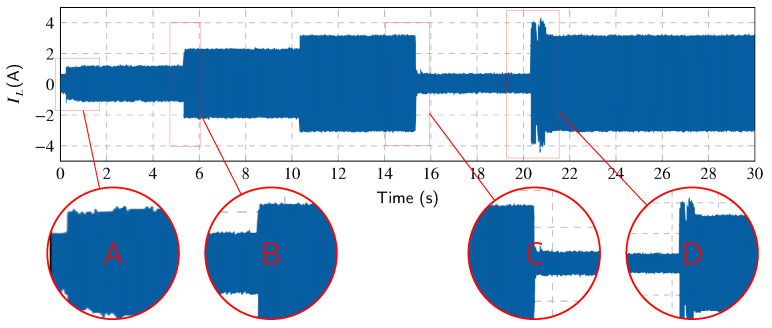
Load current $I_{L}$(t) waveform with highlighted transients.

**Figure 9 sensors-23-08494-f009:**
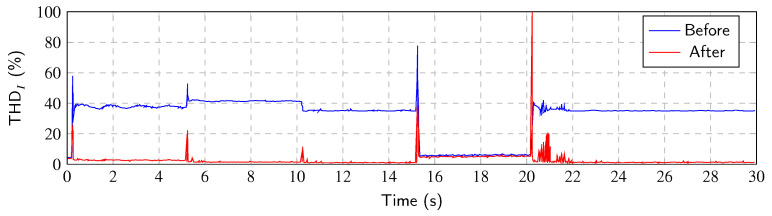
THD${}_{I}$ value of every period before and after the ideal harmonic mitigation.

**Figure 10 sensors-23-08494-f010:**
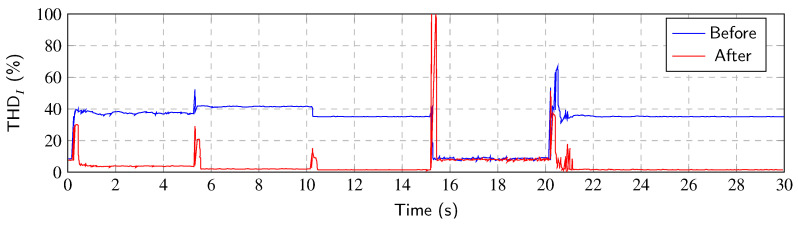
THD${}_{I}$ value of every period before and after the ideal harmonic mitigation with 2$T_{W}$ delay incorporated.

**Figure 11 sensors-23-08494-f011:**
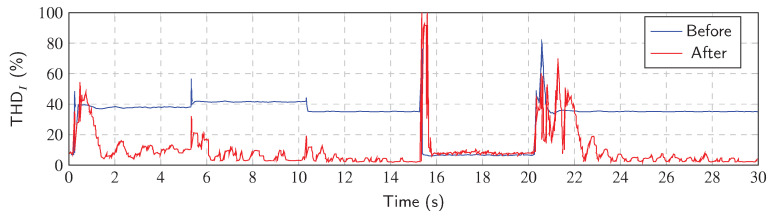
THD${}_{I}$ value of every period after Notch–LMS application.

**Figure 12 sensors-23-08494-f012:**
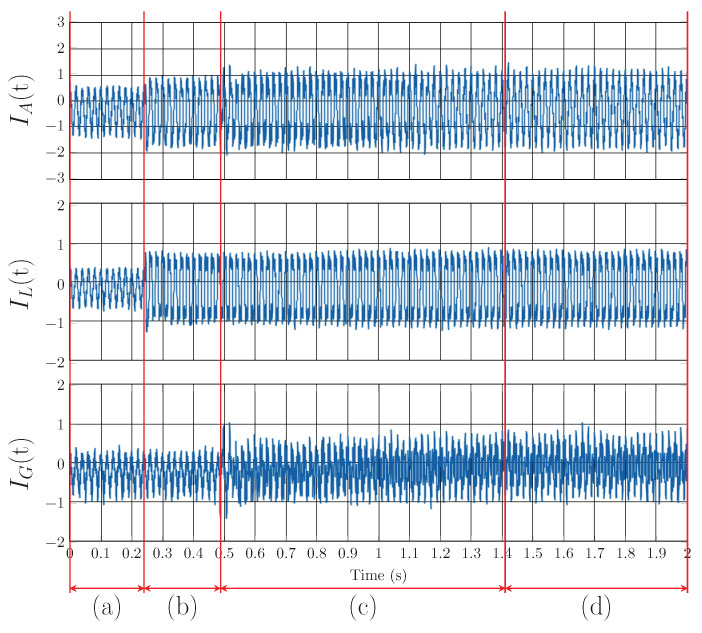
Notch–LMS application on A phase, transient *A*: (**a**) change from standby mode to run mode, (**b**) compensation delay, (**c**) adaptation time of adaptive filter and PLL, (**d**) steady state of the adaptive filter and PLL.

**Figure 13 sensors-23-08494-f013:**
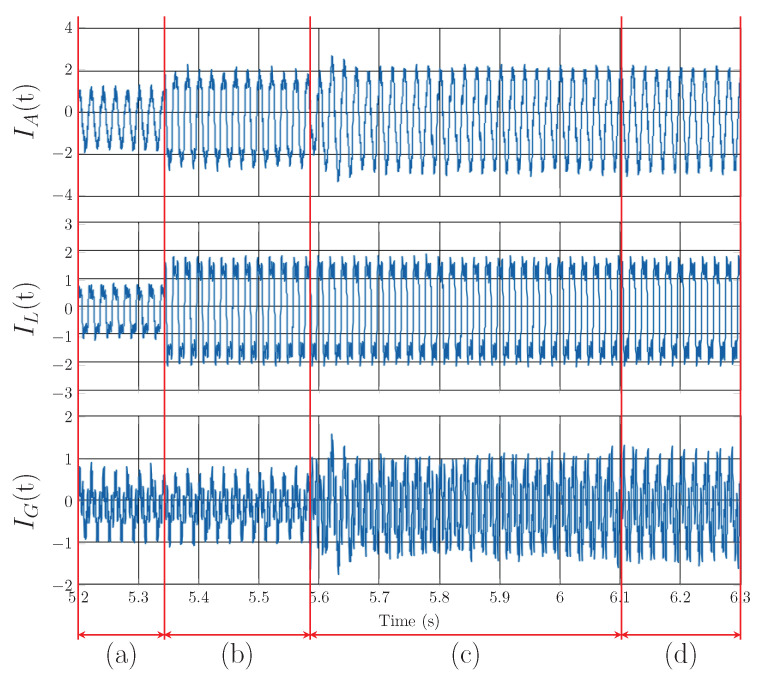
Notch–LMS application on A phase, transient *B*: (**a**) change from lower to higher current, (**b**) compensation delay, (**c**) adaptation time of adaptive filter and PLL, (**d**) steady state of the adaptive filter and PLL.

**Figure 14 sensors-23-08494-f014:**
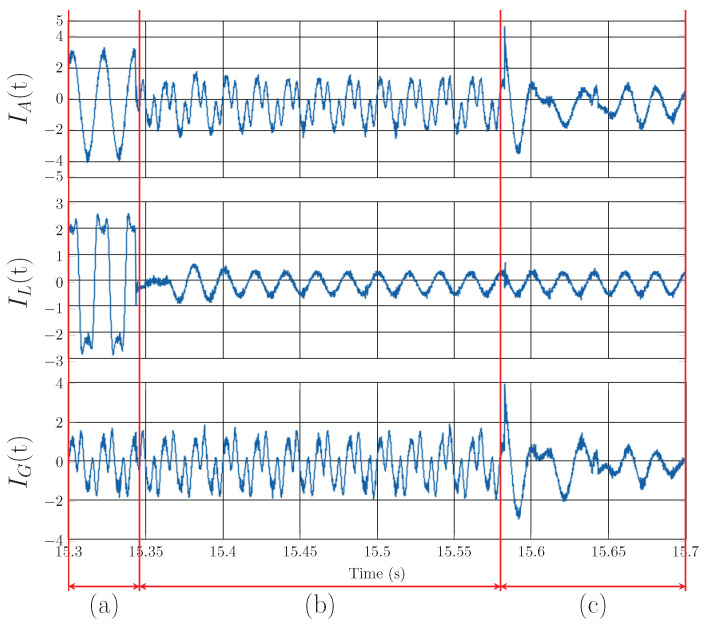
Notch–LMS application on A phase, transient *C*: (**a**) change from turned on load to standby mode, (**b**) compensation delay, (**c**) steady state of the adaptive filter and PLL.

**Figure 15 sensors-23-08494-f015:**
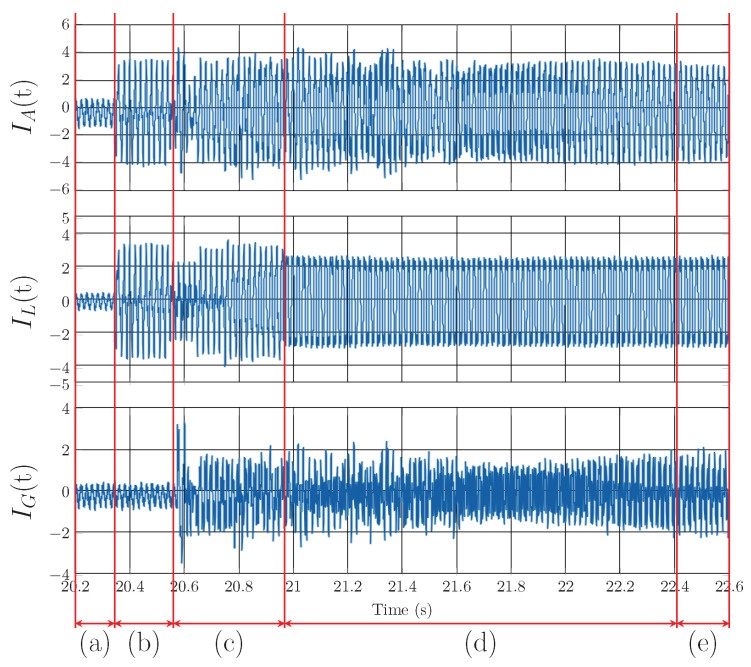
Notch–LMS application on A phase, transient *D*: (**a**) change from standby mode to higher current, (**b**) compensation delay, (**b**,**c**) adaptation time of PLL for load, (**d**) adaptation time of adaptive filter and PLL, (**e**) steady state of the adaptive filter and PLL.

**Figure 16 sensors-23-08494-f016:**
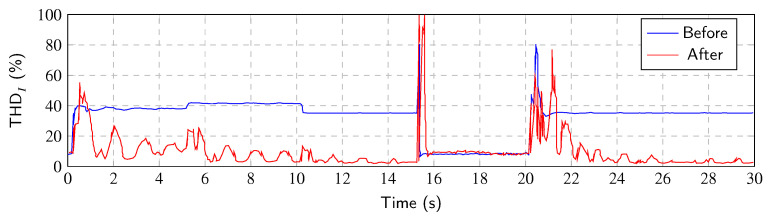
THD${}_{I}$ value of every period after Notch–RLS application.

**Figure 17 sensors-23-08494-f017:**
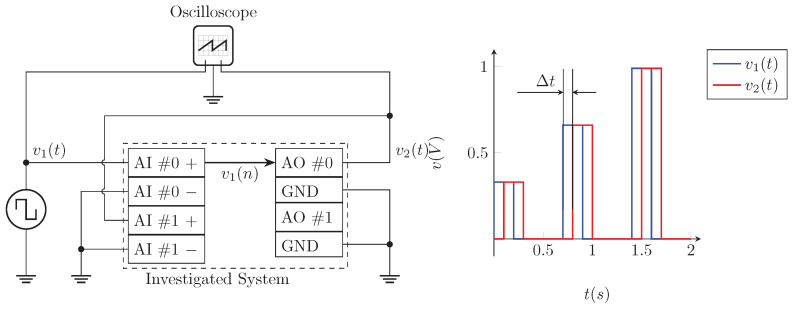
Delay measurement schematic and principle.

**Figure 18 sensors-23-08494-f018:**
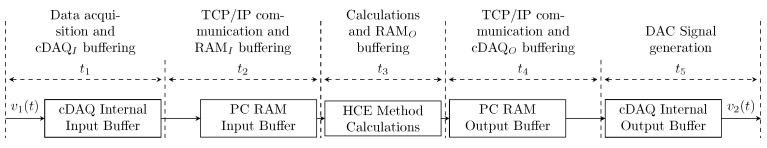
Example of cDAQ timing operations.

**Figure 19 sensors-23-08494-f019:**
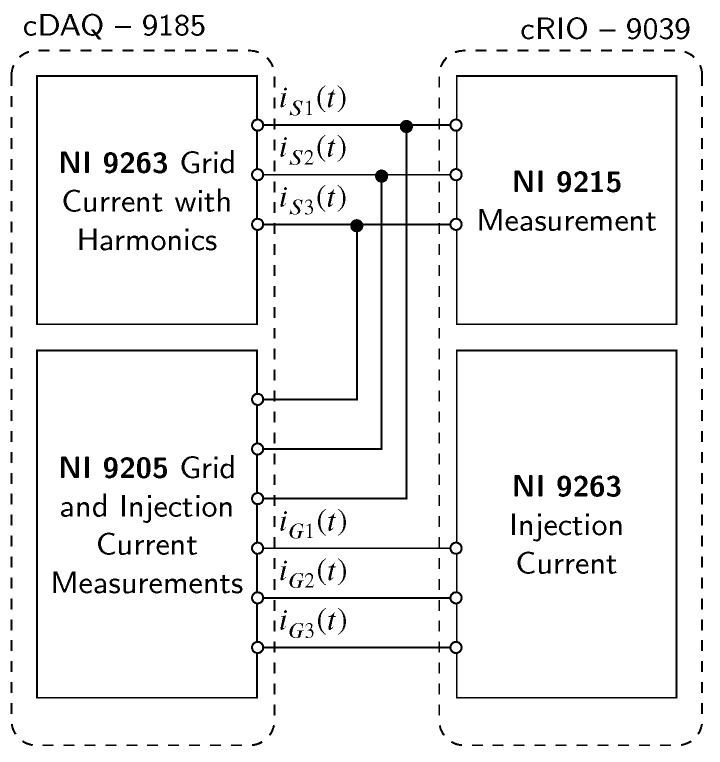
cRIO-9039 SAPF schematic and principle.

**Figure 20 sensors-23-08494-f020:**
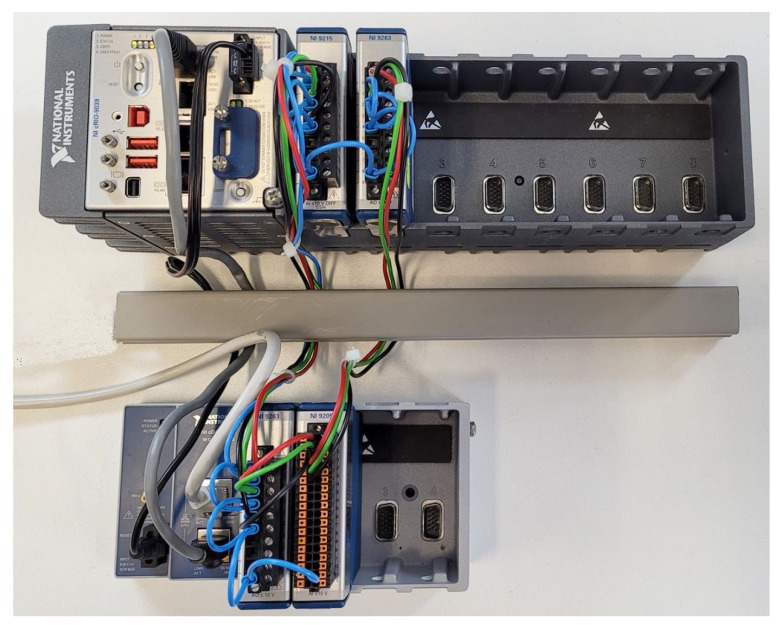
cRIO-9039 experimental SAPF setup.

**Figure 21 sensors-23-08494-f021:**
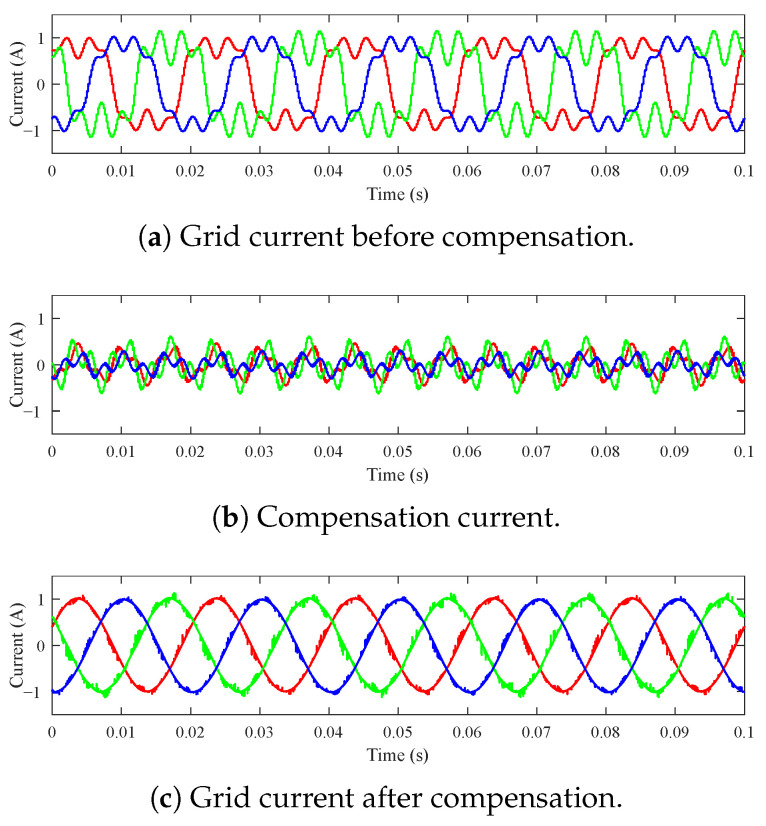
Harmonic current compensation experiment conducted on cRIO-9039 platform.

**Table 1 sensors-23-08494-t001:** Example of the scenario in the form of a .csv file that can be executed.

Index	Phase	Fundamental Current	Harmonics Definition							Time (s)
		Amplitude (A)	Index	Ratio (%)	Index	Ratio (%)	⋯	Index	Ratio (%)	
*1*	*1*	2	3	10	5	2	⋯	15	0.5	5
	*2*	0.75	3	8	5	2	⋯	15	0	
	*3*	0.5	3	10	7	5	⋯	21	0.1	
*2*	*1*	2	3	10	5	0	⋯	9	7	0.04
	*2*	0.75	3	5	5	0.5	⋯	9	7	
	*3*	0.5	3	3	7	1	⋯	9	7	
⋮	⋮	⋮	⋮	⋮	⋮	⋮	⋱	⋮	⋮	⋮
*N*	*1*	-	-	-	-	-	⋯	-	-	-
	*2*	-	-	-	-	-	⋯	-	-	
	*3*	-	-	-	-	-	⋯	-	-	

**Table 2 sensors-23-08494-t002:** Accuracy of analogue measuring modules *NI-9215* and *NI-9263*.

Type of Module	Measurement Conditions		Percent of Reading (Gain Error)	Percent of Range (Offset Error)
NI-9215	Calibrated	Maximum (−40 ${}^{\circ }$C to 70 ${}^{\circ }$C)	0.2%	0.082%
		Typical (23 ± 5 ${}^{\circ }$C)	0.02%	0.014%
	Uncalibrated	Maximum (−40 to 70 ${}^{\circ }$C)	1.05%	0.82%
		Typical (23 ± 5 ${}^{\circ }$C)	0.6%	0.38%
NI-9263	Calibrated	Maximum (−40 to 70 ${}^{\circ }$C)	0.35%	0.75%
		Typical (23 ± 5 ${}^{\circ }$C)	0.03%	0.1%
	Uncalibrated	Maximum (−40 to 70 ${}^{\circ }$C)	2.2%	1.7%
		Typical (23 ± 5 ${}^{\circ }$C)	0.3%	0.25%

**Table 3 sensors-23-08494-t003:** Output voltage and current parameters of the OMICRON CMC 256plus calibrator.

Output	Range	Typical Accuracy	Resolution	Phase (50 Hz)
Voltage	0 …300 $V_{rms}$	<0.015% reading + 0.005% range	10 mV	<0.02${}^{\circ }$ guar.
Current	0 …12.5 $A_{rms}$	<0.01% reading + 0.005% range	50 μA	<0.02${}^{\circ }$ guar.

**Table 4 sensors-23-08494-t004:** Notch-LMS algorithm function verification. Comparison of various μ settings on THD${}_{I}$ improvement.

Method	μ-Convergence Constant (-)	Avg. Rel. THD${}_{I}$ Improv. (%)
Ideal	-	29.281
Ideal + delay	-	27.473
LMS	$1\cdot 10^{-15}$	19.669
	$1\cdot 10^{-14}$	19.412
	$1\cdot 10^{-13}$	21.345
	$1\cdot 10^{-12}$	19.276
	$1\cdot 10^{-11}$	21.144
	$1\cdot 10^{-10}$	20.612
	$1\cdot 10^{-9}$	18.792
	$1\cdot 10^{-8}$	21.230
	$1\cdot 10^{-7}$	19.494
	** $1\cdot 10^{-6}$ **	**22.073**
	$1\cdot 10^{-5}$	21.544
	$1\cdot 10^{-4}$	21.423
	$1\cdot 10^{-3}$	21.872
	$1\cdot 10^{-2}$	21.624
	$1\cdot 10^{-1}$	17.011

**Table 5 sensors-23-08494-t005:** Notch-RLS algorithm function verification. Comparison of various forgetting factor settings on THD${}_{I}$ improvement.

Method	Forgetting Factor (-)	Avg. Rel. THD${}_{I}$ Improv. (%)
Ideal	-	29.281
Ideal + delay	-	27.473
RLS	0.90000	9.205
	0.99000	21.545
	0.99900	22.207
	**0.99990**	**22.346**
	0.99999	21.773
	1	21.620

**Table 6 sensors-23-08494-t006:** Effect of sample rate settings on system latency.

$f_{S}$ (kS)	1	2.5	5	10	15	25	35	40	45	50	51.02	52	52<
**Latency** (μs)	1320	520	430	216	100	138	60	68	46	46	70	Error	Error

**Table 7 sensors-23-08494-t007:** Comparison of cDAQ9185 system delay for $T_{W}=100$ ms, various $f_{S}$ and minimized RAM output buffer length.

$f_{S}$ (kHz)	62.5	50	40	30	20	15	9	5	4.5	1
Number of samples (kS)	6.25	5	4	3	2	1.5	0.9	0.5	0.45	0.1
A${}_{O}$ buff. length (S)	12,000	7400	5000	3500	2650	2300	1200	750	600	200
A${}_{O}$ buff. delay (ms)	192	148	125	117	133	153	133	150	133	200
Expected min. delay (ms)	292	248	225	217	233	253	233	250	233	300
Measured delay (ms)	269	230	187	200	214	225	199	215	202	264

**Table 8 sensors-23-08494-t008:** Comparison of cDAQ9185 system delay for decreasing $T_{W}$, $f_{S}=5$ kHz and minimized RAM output buffer length.

$T_{W}$ (ms)	100	90	80	70	60	50	40	30
Number of samples (S)	500	450	400	350	300	250	200	150
A${}_{O}$ buff. length (S)	750	570	620	550	400	350	370	410
A${}_{O}$ buff. delay (ms)	150	114	124	110	80	70	74	82
Expected min. delay (ms)	350	294	284	250	200	170	154	142
Measured delay (ms)	215	176	227	187	144	200	140	148

**Table 9 sensors-23-08494-t009:** Settings and results of the experiment.

Notch–LMS Filter			
FCU $f_{S}$ (kS)	52		
Phase	1	2	3
μ (-)	0.00099451	0.00099451	0.00099451
THD${}_{I}$ (%)	33.48	42.11	21.05
**Filtration results**			
THD${}_{I}$ (%)	0.56	1.19	0.47
ΔTHD${}_{i}$ (%)	32.92	40.92	20.58
δTHD${}_{i}$ (%)	98.32	97.18	97.77

## Data Availability

Not applicable.

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
