# Peer review of "Instrumentation for Verification of Shunt Active Power Filter Algorithms"

_sensors, 2023, doi:10.3390/s23208494_

Round 1
Reviewer 1 Report
The overall intention of the paper (sensors-2549669) entitled "Instrumentation for Verification of Shunt Active Power Filter Algorithms" by Jan Baros et al., is good and the topic is relevant with a lot of potentials. In this paper the authors present a design of a comprehensive system for testing and verifying shunt active power filter control methods. The functionality of the system is confirmed on real hardware platform using Least Mean Squares and Recursive Least Squares algorithms. Overall, the purpose of the paper is well stated in the introduction and the crucial findings are clearly and logically explained.
In my opinion, the data quality in this work is good. I think that the paper could be possibly accepted for the publication Sensor journal. Some additional points the authors should take into consideration.
Point 1: The conclusion in its current form is too wordy and seems highly unprofessional and needs to be reframed to suit the wide audience of the journal. Please focus only on key findings. It would help if you condensed it.
Point 2: The authors should check the manuscript carefully and the English needs to be further improved. There is lots of not connecting sentences.
The overall intention of the paper (sensors-2549669) entitled "Instrumentation for Verification of Shunt Active Power Filter Algorithms" by Jan Baros et al., is good and the topic is relevant with a lot of potentials. In this paper the authors present a design of a comprehensive system for testing and verifying shunt active power filter control methods. The functionality of the system is confirmed on real hardware platform using Least Mean Squares and Recursive Least Squares algorithms. Overall, the purpose of the paper is well stated in the introduction and the crucial findings are clearly and logically explained.
In my opinion, the data quality in this work is good. I think that the paper could be possibly accepted for the publication Sensor journal. Some additional points the authors should take into consideration.
Point 1: The conclusion in its current form is too wordy and seems highly unprofessional and needs to be reframed to suit the wide audience of the journal. Please focus only on key findings. It would help if you condensed it.
Point 2: The authors should check the manuscript carefully and the English needs to be further improved. There is lots of not connecting sentences.
Author Response
Reviewer 1
Dear Reviewer 1,
Thank you very much for all your constructive comments. Let us present our notes with explanations and adjustments. Every change/addition is marked by red color.
Comments and Suggestions for Authors
The overall intention of the paper (sensors-2549669) entitled "Instrumentation for Verification of Shunt Active Power Filter Algorithms" by Jan Baros et al., is good and the topic is relevant with a lot of potentials. In this paper the authors present a design of a comprehensive system for testing and verifying shunt active power filter control methods. The functionality of the system is confirmed on real hardware platform using Least Mean Squares and Recursive Least Squares algorithms. Overall, the purpose of the paper is well stated in the introduction and the crucial findings are clearly and logically explained.
In my opinion, the data quality in this work is good. I think that the paper could be possibly accepted for the publication Sensor journal. Some additional points the authors should take into consideration.
- The conclusion in its current form is too wordy and seems highly unprofessional and needs to be reframed to suit the wide audience of the journal. Please focus only on key findings. It would help if you condensed it.
Author response: Thank you for the suggestion to improve the conclusion. We tried to modify it, since it was also requested by another reviewer.
Authors’ action: Conclusion has been improved based on reviewer recommendations.
- The authors should check the manuscript carefully and the English needs to be further improved. There is lots of not connecting sentences.
Author response: We have thoroughly reviewed the English in the article and tried to improve it. We tried to edit the sentences that didn't have the right structure. If, even after the review and eventual acceptance of the article, it was of insufficient quality, we are willing to use the paid English proofreading services.
Authors’ action: The English in the article has been double-checked and improved.
Reviewer 2 Report
Dear authors, thank you for considering the journal to submit your paper. I believe that your work has some potential in its area, but it requires substantial improvement in various aspects to ensure its publication. You are requested to attend to the following comments in order to correctly visualize the importance of your work in the application area.
In the description of the components used, it is mentioned that the idea is to degrade the supply voltage and the inclusion of various inductors is presented. Here it is necessary to include additional data to be able to correctly visualize the effect of the inductors on the voltage. In this case, it is requested to include the real impedance value of the contemplated grid, that is, what is the impedance value in the supply with respect to the impedance made up of the loads.
It is mentioned that a PID control is used for regulation in the DC link, but the gain parameters used and their way of determination are not mentioned. Due to this, it is not possible to accurately visualize if the presented results still admit a certain degree of improvement if other parameters were considered in the controller. It is requested to include the complete parameters used in the controller and to mention under what aspects (rise time, settling time, maximum overshoot, error, etc.) it was determined that the chosen values ​​are optimal for the application made.
When the Notch filter equations are established, it is necessary to mention what is the minimum value of error considered to obtain an acceptable value of convergence.
It is requested to explain a little more in detail the considerations carried out for what is shown in Figure 7. In this case, explain what happens in standby mode. For the above, explain which harmonics intervene in this state and what is their origin, since the presence of distortion is noted.
As a last observation that needs to be answered, I have the following: When presenting the development of a test prototype for various algorithms, it is necessary to establish whether the developed prototype complies with various standards. In this case, it is necessary to validate and check if the stages as a whole and their components meet basic measurement, sampling, acquisition, latency, error, execution speed standards, etc. For this observation, it is requested to include a summary table with the various parameters that intervene in the precision of the application and a technical explanation of why these values ​​are considered adequate for the tests to be carried out. The requested parameters are: measurement error deviation (currents and voltages), sampling speed, PLL convergence speed, power capacity and converter stabilization time. Based on these data, issue a discussion that the values ​​obtained represent the optimal conditions for testing the highly varied control algorithms reported in the literature.
Author Response
Reviewer 2
Dear Reviewer 2,
Thank you very much for all your constructive comments. Let us present our notes with explanations and adjustments. Every change/addition is marked by red color.
Dear authors, thank you for considering the journal to submit your paper. I believe that your work has some potential in its area, but it requires substantial improvement in various aspects to ensure its publication. You are requested to attend to the following comments in order to correctly visualize the importance of your work in the application area.
- In the description of the components used, it is mentioned that the idea is to degrade the supply voltage and the inclusion of various inductors is presented. Here it is necessary to include additional data to be able to correctly visualize the effect of the inductors on the voltage. In this case, it is requested to include the real impedance value of the contemplated grid, that is, what is the impedance value in the supply with respect to the impedance made up of the loads.
Author response: Thank you for your comment on the article. Your comment confused us a bit and made us think about the whole issue further – your issues were discussed by the designer of the platform itself. With due respect, we feel it is unnecessary to include the impedance information in this article, as impedance is frequency dependent and will retain different values for each harmonic. In terms of filtering issues, the load is uniquely defined by the spectrum of the drawn current. Each harmonic component of the current causes a drop in the corresponding voltage component on the series inductance of the supply network. This voltage drop is proportional to the series inductance and frequency of the harmonic component and affects the final load voltage. We have measured the series inductance of the power network connected to the CICS workspace and added it to the text.
Authors’ action: We have added text in the section 2.3.
- It is mentioned that a PID control is used for regulation in the DC link, but the gain parameters used and their way of determination are not mentioned. Due to this, it is not possible to accurately visualize if the presented results still admit a certain degree of improvement if other parameters were considered in the controller. It is requested to include the complete parameters used in the controller and to mention under what aspects (rise time, settling time, maximum overshoot, error, etc.) it was determined that the chosen values are optimal for the application made.
Author response: Thanks for the comment about the PID controller in the DC-link. We asked the CICS injection source manufacturer for this information. The manufacturer claim that they have inserted their own custom developed module into the CICS platform, which takes care of the given PI regulation. As users of the experimental CICS platform, we are not able to describe further, more detailed, information about obtaining the optimal PID controller parameters for a given application.
Authors’ action: In section 2.1, we have added text with additional information regarding the PID controller in the DC-link, which was provided to us by the CICS manufacturer.
- When the Notch filter equations are established, it is necessary to mention what is the minimum value of error considered to obtain an acceptable value of convergence.
Author response: A great influence on the functionality of the used adaptive algorithms Notch LMS and Notch RLS is the correct setting of their input parameters. As far as Notch LMS is concerned, this algorithm needs to correctly set the convergence constant µ. For Notch RLS, the forgetting factor λ is the most important parameter. The filter order for both of these algorithms is fixed to 2. With the correct setting of the mentioned parameters, the minimum value of error is achieved. The results of the effect of setting the mentioned parameters can be seen in Table 4 and Table 5.
However, the problem is that it may be necessary to set the parameters differently for different tasks or signals. Therefore, it can be concluded that there is no universal value for these parameters and the user is usually forced to set them empirically according to experience. Another solution that is offered is the use of optimization techniques that would automatically set the aforementioned parameters and therefore make the system work without the added error of the human factor. It should be emphasized that most optimization techniques are computationally intensive, but due to the implementation of the algorithms on the FPGA, the speed of computation is greatly accelerated.
Authors’ action: The mentioned text, which is a reply to the reviewer, has been added to the discussion of the article.
- It is requested to explain a little more in detail the considerations carried out for what is shown in Figure 8. In this case, explain what happens in standby mode. For the above, explain which harmonics intervene in this state and what is their origin, since the presence of distortion is noted.
Author response: Thank you for your interesting question. Of course, the CICS platform draws current in standby mode to power the control electronics and network elements. We have performed a verification of what current is drawn in standby mode and which harmonic components are included. We hope that the description is sufficient.
Authors’ action: We have added text in the first paragraph of section 5.
- As a last observation that needs to be answered, I have the following: When presenting the development of a test prototype for various algorithms, it is necessary to establish whether the developed prototype complies with various standards. In this case, it is necessary to validate and check if the stages as a whole and their components meet basic measurement, sampling, acquisition, latency, error, execution speed standards, etc. For this observation, it is requested to include a summary table with the various parameters that intervene in the precision of the application and a technical explanation of why these values are considered adequate for the tests to be carried out. The requested parameters are: measurement error deviation (currents and voltages), sampling speed, PLL convergence speed, power capacity and converter stabilization time. Based on these data, issue a discussion that the values obtained represent the optimal conditions for testing the highly varied control algorithms reported in the literature.
Author response: Thank you for your comment on the article. We agree that we need to verify the accuracy for voltage and all current measurements. Unfortunately, the CICS prototype was partially designed by external subject and has their custom components. However, the control unit is the cDAQ chassis with the measurement modules - their parameters and accuracies are in Table 2. According to your comment we have added section 3, which deals with the adjustment and calibration of the measurement chain. The actual adjustment and calibration was done on NI-9215 modules, which ensure correct measurement of voltages and currents. It is not possible to represent the measured data during adjustment and calibration due to the large volume of data (calibration of voltages and currents in 5 parts of the range and also in the frequency domain (2nd - 50th harmonic) on each part of the range). However, after adjustment (finding the calibration constants) and subsequent calibration of the measuring chain, it was verified that the measuring system easily meets the requirements set by the standard for measuring and testing technology 61000-4-30. The sampling rate used during the experiment was 62.5KS/s, but the NI-9215 and NI-9263 simultaneous modules allow sampling rates up to 100KS/s. Both types of modules have a 16-bit converter. The settling time of the NI-9215 ADC AI measurement module with a 20V step (full range) is 15us. The settling time for the DAC AO of the NI-9263 full range measurement module is 20us. Latencies were verified for 3 different controller types and variations of several TW time window lengths and sampling rates. The results are described and clearly represented in the tables in the discussion, followed by an experimental verification that the cRIO indeed provides a minimum system response.
We have attempted to respond to your comment to the best of our ability. It is also described in the Discussion that it is not possible to test the very diverse control algorithms with the cDAQ controller because of latencies in the tens to hundreds of milliseconds. For this reason, the CICS experimental platform will be extended with the cRIO control unit to ensure that the system is able to generate the injected current with minimal latency (units of microseconds). The cDAQ measurement unit fulfils all the conditions imposed on the measurement and data representation and is an integral part of the system for testing the very diverse control algorithms reported in the literature.
Authors’ action: We have added section 3 (CICS Measurement Part Accuracy and Calibration) about adjustment and calibration.
Reviewer 3 Report
1. The introduction fails to sufficiently capture the reader's interest in the topic of instrumentation for SAPF algorithm verification. It lacks a compelling rationale for the importance of this area of research.
2. The paper lacks a coherent structure, making it difficult to follow the flow of ideas. The lack of subheadings and transitions between sections hinders the reader's ability to navigate through the content.
3. The paper is burdened with technical jargon that might alienate readers who are not intimately familiar with power electronics and SAPF concepts. Simplifying the language and providing explanations for key terms would make the paper more accessible to a wider audience.
4. The paper misses a valuable opportunity to illustrate the real-world application of the discussed instrumentation. Including case studies or examples where the described instruments have been used to verify SAPF algorithms would add practical insights and relevance.
5. The paper neglects to address the challenges and limitations associated with instrumentation for SAPF algorithm verification. Failing to discuss issues such as noise, calibration, and interference undermines the paper's practical relevance.
6. The conclusion feels abrupt and fails to effectively summarize the key takeaways from the paper. It should not only recap the main points but also reiterate the significance of the discussed instrumentation for SAPF research.
7. The writing style needs improvement. Sentences are convoluted and lack a clear structure. The lack of coherence makes it challenging to follow the author's train of thought.
N/A
Author Response
Reviewer 3
Dear Reviewer 3,
Thank you very much for all your constructive comments. Let us present our notes with explanations and adjustments. Every change/addition is marked by red color.
- The introduction fails to sufficiently capture the reader's interest in the topic of instrumentation for SAPF algorithm verification. It lacks a compelling rationale for the importance of this area of research.
Author response: Thank you for your comment regarding the introduction. We agree with your comment and have tried to give the reader a sense of the motivation and overall importance of this research in the introduction. We also discarded irrelevant information from the introduction.
Authors’ action: The introduction of the article has been revised and expanded according to this comment.
- The paper lacks a coherent structure, making it difficult to follow the flow of ideas. The lack of subheadings and transitions between sections hinders the reader's ability to navigate through the content.
Author response: We agree that the structure of the article needs imptovement. We have added to the article a new section 3 (CICS Measurement Part Accuracy and Calibration). Consequently, we have tried to adjust the structure to make it easier for the reader to understand.
Authors’ action: We have improved the structure of the article.
- The paper is burdened with technical jargon that might alienate readers who are not intimately familiar with power electronics and SAPF concepts. Simplifying the language and providing explanations for key terms would make the paper more accessible to a wider audience.
Author response: Thank you for the comment. In our article we used common abbreviations with explanations or technical terms. However, to be sure, we have double-checked the article and edited some terminology. We tried to simplify the writing language and add explanations.
Authors’ action: The article has been double-checked and edited for terminology.
- The paper misses a valuable opportunity to illustrate the real-world application of the discussed instrumentation. Including case studies or examples where the described instruments have been used to verify SAPF algorithms would add practical insights and relevance.
Author response: Thank you for your comment. We would like to add case studies and examples to the paper that describe the proposed system used to verify the SAPF algorithms, but we have not found any comprehensive, designed and constructed real system with controlled current generators and synthetic loads. Many of the publications found dealing with SAPF algorithms describe verification using real experiments on a real SAPF or in a SW simulation environment. Many of the methods have been subjected to comparative experiments, but these did not follow a uniform methodology or a uniform HW. The conditions under which the methods were investigated and evaluated varied from publication to publication. Therefore, it is not possible to objectively compare individual experiments by different authors.
This was also our motivation to create the presented unique CICS experimental platform for quality verification of SAPF control algorithms. This platform is important for SAPF research because it allows us to verify all kinds of harmonic extraction algorithms under the same and different states of the power grid using controllable loads.
Authors’ action: We have expanded the text at the introduction of the article, which partially resolved your comment.
- The paper neglects to address the challenges and limitations associated with instrumentation for SAPF algorithm verification. Failing to discuss issues such as noise, calibration, and interference undermines the paper's practical relevance.
Author response: Thank you for your comments. This paper presents a novel design and validation of a unique, custom-built, open-architecture experimental platform based on virtual instrumentation for R&D purposes. CICS contains power electronics (injected current generators, programmable loads) that load the grid with harmonically biased current in standby and run mode. Per a comment from a previous reviewer, verification of how the IL(t) current is affected by disturbances when CICS is turned on with the load in standby mode has been added.
One limitation occurred during experimental verification of the SAPF algorithms. In the discussion of this paper, it is described and also experimentally verified that due to the high latency of the cDAQ measurement system, the CICS platform needs to be augmented with the cRIO platform to provide real-time control and generation of injected current with latency on the order of units microseconds.
Since CICS contains signal conditioning modules and the cDAQ measurement system based on virtual instrumentation, the measurement chain adjustment & calibration of the CICS experimental platform had to be performed before the experiment was performed. We have extended the paper with section 3 (CICS Measurement Part Accuracy and Calibration) we consider important and could improve the practical emphasis of the paper.
Authors’ action: We have added section 3 (CICS Measurement Part Accuracy and Calibration) about calibration and texts about verification of how the IL(t) current is affected by disturbances into section 5 (Experimental Verification).
- The conclusion feels abrupt and fails to effectively summarize the key takeaways from the paper. It should not only recap the main points but also reiterate the significance of the discussed instrumentation for SAPF research.
Author response: Thank you for the suggestion to improve the conclusion. Since it was also mentioned by other reviewer, the whole conclusion chapter was vastly modified.
Authors’ action: Conclusion has been improved based on reviewer recommendations.
- The writing style needs improvement. Sentences are convoluted and lack a clear structure. The lack of coherence makes it challenging to follow the author's train of thought.
Author response: We have thoroughly reviewed the English in the article and tried to improve it. We tried to edit the sentences that didn't have the right structure. If, even after the review and eventual acceptance of the article, it was of insufficient quality, we are willing to use the paid English proofreading services.
Authors’ action: The English in the article has been double-checked.
Round 2
Reviewer 3 Report
No more comments.
N/A